# Oxidative Stress Markers in Multiple Sclerosis

**DOI:** 10.3390/ijms25126289

**Published:** 2024-06-07

**Authors:** Félix Javier Jiménez-Jiménez, Hortensia Alonso-Navarro, Paula Salgado-Cámara, Elena García-Martín, José A. G. Agúndez

**Affiliations:** 1Section of Neurology, Hospital Universitario del Sureste, Arganda del Rey, E-28500 Madrid, Spain; hortalon@yahoo.es (H.A.-N.); paula.salgado.camara@gmail.com (P.S.-C.); 2University Institute of Molecular Pathology Biomarkers, Universidad de Extremadura, E-10071 Cáceres, Spain; elenag@unex.es (E.G.-M.); jagundez@unex.es (J.A.G.A.)

**Keywords:** multiple sclerosis, pathogenesis, risk factors, oxidative stress, biological markers, animal models

## Abstract

The pathogenesis of multiple sclerosis (MS) is not completely understood, but genetic factors, autoimmunity, inflammation, demyelination, and neurodegeneration seem to play a significant role. Data from analyses of central nervous system autopsy material from patients diagnosed with multiple sclerosis, as well as from studies in the main experimental model of multiple sclerosis, experimental autoimmune encephalomyelitis (EAE), suggest the possibility of a role of oxidative stress as well. In this narrative review, we summarize the main data from studies reported on oxidative stress markers in patients diagnosed with MS and in experimental models of MS (mainly EAE), and case–control association studies on the possible association of candidate genes related to oxidative stress with risk for MS. Most studies have shown an increase in markers of oxidative stress, a decrease in antioxidant substances, or both, with cerebrospinal fluid and serum/plasma malonyl-dialdehyde being the most reliable markers. This topic requires further prospective, multicenter studies with a long-term follow-up period involving a large number of patients with MS and controls.

## 1. Introduction

Multiple sclerosis (MS), which is characterized mainly by inflammation, demyelination, and neuronal degeneration, is considered to be a chronic autoimmune disease with a genetic predisposition affecting the central nervous system. To date, at least 200 loci with genome-wide significance have been associated with the risk for MS through genome-wide association studies (GWAS) [1,2]. Most of the described associations, however, show a modest odds ratio (OR) and explain close to half of its heritability [1,2], HLA (in particular, the HLA-DRB1*15:01 haplotype) being the only one that has shown a strong association with MS risk [1]. It has been suggested that, together with genetic predisposition, some environmental factors, gene–environment, and environment–environment interactions, including smoking, infections (mainly Epstein–Barr virus seropositivity or exposure), low sun exposure/low vitamin D levels, and obesity may be related to the etiopathogenesis of MS and with MS onset and progression [3,4,5]. Since it has been suggested that oxidative stress is closely related to inflammation (for example, in inflammatory conditions, immune cells can liberate reactive oxidant substances leading to oxidative stress and, on the other hand, the oxidative damage produced by free radicals can induce an inflammatory response through the Toll-like receptors and inflammasomes) [6,7], with MS being a prototype of inflammatory diseases, oxidative stress could also play a role in the etiopathogenesis of MS. Figure 1 depicts the possible interaction between the different mechanisms proposed in the etiopathogenesis of MS, including oxidative stress.

The term “oxidative stress” designates the imbalance between the production of reactive oxygen species (ROS) and the ability of a biological system to neutralize intermediate reagents or to repair the resulting damage. Biomarkers of oxidative stress can be divided into molecules modified by their interaction with ROS or free radicals derived from nitrogen (RNS) and into molecules of the antioxidant system in response to an increase in redox stress. These include lipid peroxidation, protein oxidation, DNA oxidation markers, enzymes or protein with antioxidant actions, other prooxidant and antioxidant substances, and global markers of oxidative processes, such as the total oxidant status/capacity (TOS/TOC), total antioxidant status/capacity (TAS/TAC), and oxidative stress index (OSI).

This narrative review aims to analyze the results of published studies on the possible role of oxidative stress in multiple sclerosis, mainly those related to oxidative stress markers in different tissues from patients diagnosed with MS, but also case–control association studies on the possible association of candidate genes related to oxidative stress with risk for MS, and studies showing the presence of oxidative stress in experimental models of multiple sclerosis. To this end, we performed a PubMed Database search from 1966 to 28 December 2023, crossing the terms “multiple sclerosis” and “oxidative stress”. The search retrieved 1672 references that were manually selected to include only those strictly related to the topic (a total of 201 references).

## 2. Oxidative Stress Markers in Patients with Multiple Sclerosis

### 2.1. Oxidative Stress Markers in the Brain and Spinal Cord

The results of studies on oxidative markers in the brains or spinal cord of patients with MS, with most of them compared to controls [8,9,10,11,12,13,14,15,16,17,18,19,20,21,22,23,24,25,26,27,28,29,30,31], are summarized in Appendix A. Many autopsy studies describe an increase in various markers of lipid peroxidation [8,9,10,11,12,13,14,15], an increase in carbonylated proteins [16], markers of DNA damage [9,11,12,17], and nitrotyrosine (a marker of nitrosative stress) [8,11] in the brains of patients diagnosed with MS, especially in active MS plaques. The enzymatic activity of superoxide-dismutase 1 and 2 is upregulated in active demyelinating lesions [10] and in cerebellar gray matter of patients with MS [11], and catalase activity is increased in active demyelinating lesions [11]. In contrast, glutathione peroxidase (GPx) [10] and catalase [10] activities are similar to those of controls in cerebellar gray matter. Iron content has been found to decrease in MS inactive lesions [12], mitochondrial protein expression is increased, and mitochondrial complex IV activity is upregulated in MS lesions [18]. Studies with proton magnetic resonance spectroscopy (^1^HMRS) found decreased glutathione in some brain regions of MS patients [19,20,21].

Several studies have also reported the upregulation of multiple enzymes and proteins involved in oxidative processes, such as NAD(P)H:quinone oxidoreductase 1 (NQO1) [22], some subunits of NADPH oxidase 2 [24], nicotinamide adenine dinucleotide phosphate oxidase 1 (NOX1) [24], nicotinamide adenine dinucleotide phosphate oxidase organizer [24], heme oxygenase 1 (HO-1) [25], myeloperoxidase (MPO) [26], metallothionein I + II [11], peroxiredoxins (PRX) 2 [23] and 5 [27], endoplasmic reticulum stress-related signaling pathway molecules [28], transcription factor NF-E2-related factor 2 (Nrf2) [29], DJ-1 protein [29], genes involved in mitochondrial protein synthesis (MRPL18, 14, 23; MRPS15, 22) [24], genes involved in adenine nucleotide translocation (SLC25A4) [24], and genes induced by oxidative stress and involved in the oxidative stress defense (UCP3, GRPEL1, TXNRD2, ISCU, AASS, ACADL, DMGDH, and CADS) [24], in brain MS lesions in comparison to control brains. On the other hand, peptidases of the 20S and 26 proteasomes [31], regulatory caps 11S α and 19S [31], and nuclear-encoded genes of the respiratory chain [24], mitochondrial DNA-encoded gene rays (ND1, ND2, ND3, ND5, ND6, COX1, and CYTB) [24], were found to be decreased or downregulated when compared to controls. Finally, brain concentrations of 20S proteasome α, β1, β2, and β5 subunits [31], calpain [31], cathepsin B [31], and mitochondrial LonP [31] have been reported to be similar in MS patients compared to controls.

### 2.2. Oxidative Stress Markers in Cerebrospinal Fluid (CSF)

Appendix A summarizes the results of studies related to markers of oxidative stress in the CSF from MS patients compared to controls [32,33,34,35,36,37,38,39,40,41,42,43,44,45,46,47,48,49,50,51,52,53,54,55,56,57,58,59,60,61,62,63,64,65,66,67]. Most of these studies found increased CSF levels of markers of lipid peroxidation, such as malonyl-dialdehyde, hydroxyalkenals, diene conjugates, 4-hydroxy-nonenal, oxidized phosphatidylcholine, and isoprostanes in patients diagnosed with MS compared to controls [13,32,33,34,35,37,38,39,40,45,46,47], with some exceptions [36]. CSF levels of certain prostaglandins were increased in patients with MS in most studies [35,41,42,43,44], but were similar to controls in others [36,37,38].

CSF protein carbonyl concentrations were increased in MS patients compared to controls in two studies [40,47], and were similar to controls in another [35]. Protein-linked neuroketals were found to be increased [35], and advanced glycoxidation end products were similar to controls [48], respectively, in two studies. Advanced oxidation protein products were increased in the CSF of MS patients compared to controls [49] and were similar for the different clinical subtypes of MS [50]. CSF levels of markers of DNA damage were increased in MS patients compared to controls [38,50]. Nitrotyrosine (a marker of nitrosative stress) was also increased in MS patients compared to controls [36].

CSF iron levels were reported to be similar in MS patients and controls [51], and transferrin was found to be decreased in MS patients with a shorter MS duration [52]. CSF concentrations of copper [51,53], ceruloplasmin [53], and matrix-metalloproteinase 9 (MMP9) [57] were increased, while ferric-reducing antioxidant power [50] and ferroxidase activity [54] were described as being similar in MS patients compared with controls. The total antioxidant status or capacity (TAS or TAC) were decreased in MS (especially in RRMS) patients compared to controls in three studies [37,43,56], increased in one study [34], and similar in another [37]. The CSF total thiol (SH) groups were decreased in MS patients [33,49]. GPx [32] and glutathione-reductase (GSSG-R) were found to be increased and aryl esterase activity was similar [58] in comparison with controls in isolated studies.

The CSF concentrations of several antioxidants, such as ascorbate [51] and the antiaging antioxidant protein Klotho [55], were decreased in MS patients. CSF levels of alpha-tocopherol were similar in MS patients and controls [59]. Two studies measuring CSF uric acid concentrations found increased levels of this antioxidant and its precursors hypoxanthine and xanthine in MS patients compared to controls [63,64], while another study described similar values of uric acid and its metabolite allantoine in MS and controls [62].

CSF levels of the excitatory amino acid L-glutamate were decreased in MS patients compared to controls [60]. CSF levels of nitric oxide (NO) metabolites were reported to be increased in MS patients compared to controls in three studies [36,42,57], and similar to those of controls in another [61]. CSF levels of trace metals involved in oxidative stress processes were the subject of a single study, which described increased lead, decreased magnesium, and similar calcium, manganese, and zinc levels in patients with primary progressive MS (PPMS) compared to those with secondary progressive MS (SPMS) and controls [51]. The CSF human serum albumin (HAS), mercaptoalbumin (HMA), and non-mercaptoalbumins 1 and 2 (HNA1 and HNA2) [65] levels were reported to be similar in MS patients and controls. CSF neutrophil gelatinase-associated lipocalin (NGAL) was increased [39] or similar [38] compared to controls in two studies by the same group. Finally, DJ-1 [66], periredoxins 2 (PRX2) [38,39], and β-site amyloid precursor protein-cleaving enzyme 1 (BACE1) [67] were increased in the CSF of patients with MS compared to controls.

### 2.3. Oxidative Stress Markers in Blood Cells

The results of studies addressing concentrations of oxidative stress markers in blood cells (erythrocytes, leukocytes, peripheral blood mononuclear cells/lymphocytes, and platelets) [68,69,70,71,72,73,74,75,76,77,78,79,80,81,82,83,84] are summarized in Appendix A. MDA/TBA/TBARS levels were described as increased in erythrocytes [68] and leukocytes [72] from patients diagnosed with MS compared with controls, and were higher in patients with a more severe disease [72]. Radical oxygen species production was increased as well in platelets from patients with SPMS compared to controls [83]. However, other lipid peroxidation markers, such as diene conjugate and fatty acid patterns of phospholipids, were reported as being similar in MS patients and controls in erythrocytes [69] and leukocytes [69,73,74]. Advanced oxidation protein products were increased in erythrocytes from patients with CIS and RRSS (being even higher in RRSS and patients with a more severe disease) [68], and in platelets from SPMS patients (higher in patients with a more severe disease) [83] in comparison with controls, 3-nitrotyrosine was found to be increased in platelets from SPMS patients compared to controls [83], and DNA damage was increased in leukocytes from patients with RRMS [75] and in PBMC from patients with MS compared to controls [81]. Global mitochondrial activity PBMC levels were similar in MS patients and controls in one study [80], while another study found a significant increase in mitochondrial respiratory chain complexes I, II, III, IV, and V [78], and another found a decreased complex IV in MS patients compared to controls [82].

Regarding antioxidant enzyme activities or levels of antioxidant substances, in patients with MS compared to controls:SOD activity was decreased in erythrocytes [68,70], leukocytes [70], and peripheral blood mononuclear cells (PBMC) [77]. Related to this finding, superoxide anion (O_2_^−^) production in MS patients was increased in PBMC [78] and in platelets [73], but was similar in leukocytes [76] from MS patients compared to controls.GPx activity was decreased in erythrocytes [69] and decreased [72] or similar [69] in leukocytes.Catalase activity was decreased [71] or similar [69] in erythrocytes, similar in leukocytes [69], and increased in PBMC [77].Myeloperoxidase activity was similar [76].Reduced (GSH) and oxidized glutathione (GSSG) concentrations were reported as being similar in erythrocytes [69] and leukocytes [69,72].Coenzyme Q_10_ concentrations were similar in erythrocytes [69] and decreased in leukocytes [69], and alpha-tocopherol levels were similar in erythrocytes [69] and leukocytes [69].The TAS was similar [73,74] and total antiradical activity was increased [73,74] in leukocytes.Thiol group concentrations were decreased [83], and NADPH oxidase (NOX-1), cytochrome c oxidase subunit 1 expression and glyceraldehyde 3-phosphate dehydrogenase activity (GADPH) were increased in platelets [83].

Finally, HO-1 [40], HsC70 [40], Hsp72 [40], and Trx concentrations [40], polyADP ribose (PAR) synthesis [81], and polyADP ribose polymerase-1 (PARP1) expression [81] were increased, HO-2 [40] and Hsp70-2 concentrations [80] were similar, and sirtuin 1 concentrations were increased [40] or similar [77] in PBMC from patients with MS compared with controls.

### 2.4. Oxidative Stress Markers in Serum/Plasma

The results of studies addressing serum and/or plasmatic levels of oxidative stress markers in patients diagnosed with MS compared to controls [33,34,36,40,44,48,49,50,51,52,53,54,56,58,59,62,63,64,65,66,69,72,73,75,85,86,87,88,89,90,91,92,93,94,95,96,97,98,99,100,101,102,103,104,105,106,107,108,109,110,111,112,113,114,115,116,117,118,119,120,121,122,123,124,125,126,127,128,129,130,131,132,133,134,135,136,137,138,139,140,141,142,143,144,145,146,147,148,149,150,151,152,153,154,155,156,157,158,159,160,161] are summarized in Appendix A.

Regarding lipid peroxidation markers, serum/plasma concentrations of MDA/TBA/TBARS were increased in MS patients compared to controls in 16 studies [33,64,72,75,85,86,87,88,90,91,92,93,94,96,97,98], but were similar in 4 studies [36,89,95,100] and decreased in 2 other studies [34,99]. Serum levels of MDA + hydroxyalkenals [101], “lipid oxidizability” [102], lipid peroxides [103], and fluorescent lipid peroxidation products [104], and lipid hydroperoxides [105,106,107,108] were increased in MS patients in most studies, and similar in others [89,103]. Serum tert-butyl hydroperoxide and hydroperoxide concentrations were reported to be similar [110,111,112,115] or increased in MS patients [109,114]. Serum HNE [40] and fatty acid patterns of phospholipids [69] levels were increased in MS patients compared to controls, while prostaglandin-F2alpha and F2-isoprostane were reported to be increased [36,116] or decreased [44].

Serum/plasma levels of protein carbonyls were described as increased in MS or RRMS patients compared to controls in nine studies [40,75,92,96,109,117,118,119,120], and similar between two groups in three other studies [108,110,115]. Advanced glycoxidation end products were normal [48,91] or increased [120], and advanced oxidation protein products were increased in MS according to eight studies [49,106,112,115,120,121,122,124], with this increase being more marked in patients with a more severe disease [115], higher ferritin levels [110], and with a trend towards a decrease during follow-up [123]. Other authors reported similar serum/plasma levels of advanced oxidation protein products in MS patients and controls (with a trend towards lower values in patients with low vitamin D levels) [111] and a lack in differences among the different evolutive types of MS [50]. Serum fructosamine levels were reported as increased in MS patients in a single study [91]. Serum/plasma levels of markers of oxidative stress/damage of DNA, such as 8-OHdG [105,118,125] and DNA single-strand breaks [99], were increased in MS patients.

Serum/plasma ferric-reducing antioxidant power (FRAP) and total-reducing antioxidant power (TRAP) were decreased [93,108,110,111,122,127] or similar [50,91,124] in MS patients compared to controls, and were similar in MS patients with different degrees of severity [107]. In comparison to controls, MS patients showed decreased [129,132,133] or similar [51,110,114,128,131,134] serum/plasma iron levels, increased ferritin levels [107,108,110,133] or those similar to controls [110], decreased [52,97] or similar [114,128] transferrin levels, higher soluble transferrin receptor levels [118,131], and similar lactoferrin levels to controls [135]. The serum/plasma total ferroxidase activity was decreased [56,136], ceruloplasmin increased [114,119,137], and copper increased [53,114,132] or similar [51,129,134] in MS compared to controls.

Serum/plasma TAS/TAC or total antiradical activity were reported to be decreased [34,77,78,87,91,94,96,102,109,114,118,130,132,138,139,141] or similar [56,73,85,89,95,97,100,126,142], TOS increased [87,97,103,118,126,129,130,132,139,140,143] or similar [103,129,140], and oxidative stress index (OSI) increased [87,97,126,139] or similar [103] in patients with MS compared to controls.

Serum/plasma total thiol group concentrations were decreased in MS patients compared to controls in most studies [33,49,75,96,109,112,122,143], with one exception [144], and were similar in patients with MS with low vs. those with high ferritin levels. Native thiol levels were found to be decreased in one study [143] and similar to those of controls in another [144]. In comparison to controls, serum plasma SOD activity from MS patients was reported to be similar [90,100,103,146], increased in MS patients [86,97,118,145], or decreased in MS patients [75,143]; total glutathione was similar [89] or increased in MS patients [118]; GSH was similar [69,89,100] or increased in MS patients [118]: GSSG was similar [68] or decreased in MS patients [118]; GSSG-reductase activity was decreased in MS patients [118,146]; GPx activity was similar [69,100,146], increased in MS patients [101], or decreased in MS patients [103,118,135]; catalase activity was increased [75,97,103] or similar [69,100]; and GST was similar [146] or decreased in MS patients [118].

Serum/plasma paraoxonase (PON1) activity was decreased in most studies in patients with MS [58,105,127,146], while others reported an increase [148] or similarity to controls [93]. Arylesterase activity in MS patients was similar to controls in three studies [123,139,146] and decreased in another [58].

Serum/plasma concentrations of coenzyme Q_10_ were reported as being decreased in MS patients compared to controls [64,69,89] or similar in MS patients and controls [151]. Similarly, four studies showed lower serum/plasma alpha-tocopherol levels in MS patients [59,69,72,75] and another showed non-significant differences when compared to controls [69]. In comparison with controls, serum/plasma gamma-tocopherol levels were decreased in MS patients in a single study [64], beta-carotene was decreased in two studies [72,152] and similar in one [64], and ascorbic acid was decreased in two studies [75,86] and similar in another two [51,72].

Many studies have addressed serum/plasma levels of NO metabolites (nitrates + nitrites) in MS patients and controls; seven showed a significant decrease [36,87,107,108,109,110,111], one showed a non-significant trend towards a decrease [61], and five others reported a significant increase [86,99,101,103,112] of these parameters in MS patients. Serum nitrotyrosine levels were increased in MS patients compared to controls [36,92,117].

Several studies have addressed serum/plasma trace metal concentrations in MS patients and controls, described in detail in Appendix A, with varying results. The most consistent findings were increased cadmium [129,132,154,155], aluminum [129,132,134], molybdenum [129,132,134], tin [129,132], zirconium [129,132], and arsenic levels in MS patients [88,93,134,155], and a lack in differences in serum/plasma levels of lead [129,132,154,155], mercury [129,132], strontium [129,130,132], vanadium [129,132,134], and wolframium [129,132].

Serum/plasma levels of uric acid were decreased in patients with MS compared to controls in three studies [97,158,159], increased in another two [63,86], and similar to controls in two more [109,156], while the serum levels of the related substances hypoxanthine, xanthine, and uridine were found to be increased [158] and allantoine was similar in MS patients compared to controls. Serum/plasma uric acid levels from MS patients were decreased in current smokers compared with non-smokers and ex-smokers [160]. Serum/plasma bilirubin levels were decreased in MS patients [97,157]. Ischemia-modified albumin was increased [140] and irisin and nesfatin-1 were decreased in MS patients compared to controls [161]. Finally, serum/plasma levels of HAS [65], HMA [65], HNA1 [65], HNA2 [65], and DJ-1 [66] from MS patients did not differ significantly from those of controls.

### 2.5. Oxidative Stress Markers in Other Fluids

Several studies found increased levels of lipid peroxidation markers [85,162], increased levels of aluminum [162], decreased levels of silicon [162], increased 8-iso-prostaglandin (PG-)F2alpha levels [163], neopterin/creatinine ratio [164], and prolyl oligopeptidase levels [165] and decreased levels of alpha2-macroglobulin [165] in urine from MS patients compared to controls (Appendix A).

Karlík et al. [91] described an increase in salivary levels of TBA/TBARS and advanced glycation end-products, decreased FRAP, and similar advanced oxidation protein products and TAS in patients with MS compared to controls (Appendix A).

## 3. Genetic Variants of Genes Related to Oxidative Stress in Patients with Multiple Sclerosis

The possible association between single nucleotide polymorphisms (SNPs) or deletions in genes related to oxidative stress and the risk of developing MS has been the subject of several case–control association studies, which are summarized in Appendix A [113,155,166,167,168,169,170,171,172,173,174,175,176,177,178,179,180]. Most of these studies found a lack of a direct association between these SNPs and multiple sclerosis, including the most common SNPs in CYP2D6 [166], GSTP1 [167,168,169], GSTM1 [113,155,167,170], GSST1 [167,170], GSTM3 [167], PON1 rs662 [173,174], PON1 rs854560 [173], NQO1 rs1800556 [176], HMOX1 2071747 [177], HMOX2 rs270363 [177], HMOX2 rs1051308 [177], NCF1 D7S2518 [178], NCF2 [178], NCF4 [178], CYBA [178], CYBB rs9330580 [178], NOS1 rs1879417 [179], NOS3 rs2070744 [180], and HNF1A-AS1 rs7953249 [181] genes.

Other authors reported a significant association for GSTM1 null polymorphism [171], GSTT1 null polymorphism [171], MPO rs2333227 [172], PON1 rs854560 [174], GLO1 rs1049346 [173], NQO1 rs1800566 [169,175], OGG1 rs1052133 [113], NCF1 D7S1870 [178], CYBB rs5963310 [178], NOS2 rs2297518 [179], CAT rs7943316 [179], and TRPP2-AS rs933151 [181] with the risk of developing MS, and a decreased risk related with SOD2 rs187947 [179] and GPX4 rs713041 [179]. Mann et al. [167] described an association between the combination of GSTM1 null polymorphism and GSTP1 rs1695 alleles and the presence of GSTM3 rs1799735 with severe disability in patients with an MS duration longer than 10 years [167]. Alexoudi et al. [169] described an interaction between GSTP1 rs1695 and NQO1 rs1800665 and the risk for MS. Finally, Agúndez et al. [178] described an association of the HMOX2 rs1051308AA genotype and rs1051308 with risk for MS in males.

## 4. Data from Experimental Models of Multiple Sclerosis

### 4.1. Lipid Peroxidation Markers

Perianes-Cachero et al. [182] described an increase in lipid peroxidation, and in SOD, GPx, GSSG-reductase activities (assessed using spectrophotometry), and a decrease in catalase activity (assessed using spectrophotometry) and GSH concentrations (assessed using a fluorometric method) in the hippocampus of 6-week-old female Lewis rats with chronic relapsing experimental autoimmune encephalomyelitis (EAE), the most important animal model of MS.

Dimitrijević et al. [183] reported an increased MDA (assessed using a colorimetric method) and superoxide anion levels, decreased GSH concentrations, decreased SOD activity (assessed using spectrophotometry), increased NOS3 and xanthine oxidase (an enzyme responsible for the synthesis of uric acid) expression (assessed using quantitative real time-polymerase chain reaction—qRT-PCR), and increased AOPP in the spinal cord from Dark Agouti rats with EAE, and increased plasma AOPP levels (assessed using high-performance size-exclusion matrix chromatography) in the same MS model.

Jhelum et al. [184] reported increased peroxidation, increased mRNA levels of several ferroptosis genes, increased nuclear receptor coactivator 4 (NCOA4) expression (assessed using qRT-PCR), decreased GPx4 activity (assessed using Western blot analysis), and decreased total glutathione (assessed using spectrophotometry) in the brain of female mice with EAE of C57BL/6. C57BL/6 OlaHSD [15] and C57BL/6 female mice with EAE [185] showed increased levels of acrolein or their metabolites (assessed using Western blot [15] or liquid chromatography/tandem mass spectrometry [185]) in the spinal cord and urine. A decrease in mRNA expression was described for the cytoplasmic isoform of GPx4 (assessed using RT-PCR) in the spinal cord from female C57BL/6 mice with EAE [186]. Smerjac and Bizzozzero [187] described increased lipid peroxidation markers (assessed by a colorimetric method) before the appearance of neurological symptoms, in the spinal cord of seven-week-old male Lewis rats with acute EAE.

### 4.2. Protein Oxidation Markers

Smerjac and Bizzozzero [187] described increased protein carbonylation and degradation (assessed using Western blotting) at the time of maximal clinical disability and a decreased glutathione concentration (assessed using spectrophotometry) in the spinal cord of seven-week-old male Lewis rats with acute EAE. The same group described an increased protein carbonylation (using the OxyBlot™ protein oxidation detection kit) within cerebellar astrocytes, which was maximal in the acute phase and decreased in the chronic phase of the disease [188], and in the spinal cord [189] of eight-week-old female C57BL/6 mice with EAE.

Castegna et al. [190] described an increased protein oxidation (assessed with OxyBlot and mass spectrometry analyses), increased glutamate/glutamine, and decreased natural antioxidant levels (assessed with liquid chromatography-tandem mass spectrometry analysis, LC-MS/MS), which paralleled disease activity in the brain of female 10–11-week-old PLSJL mice with EAE.

### 4.3. Heme Oxygenase 1 (HO-1)

Several authors have reported the increased expression of heme oxygenase 1 (HO-1) by using the Western blot analysis [184,191,192,193] and decreased expression of NADPH cytochrome P450 reductase (which is required for the catalytic activity of HO-1) expression [191] in the brain of female C57BL/6 mice [184], female SJL mice [191], pregnant Sprague-Dawley rats [192], and adult male Lewis rats [193] with EAE. An increase in HO-1 expression in the EAE brain was inhibited and exacerbated, respectively, through the coadministration of inducers of inhibitors of HO-1 [193]. HO-1 was increased in the spinal cord of the EAE rodents [192]. In addition, a significant upregulation in HO-1 and the iron storage protein ferritin (assessed with RT-PCR) in a demyelination model of mutant rats (dmv rats) was described in comparison with a hypomyelination model (mv rats) [194].

### 4.4. NAD(P)H Oxidases (NOX)

NAD(P)H oxidase (NOX) enzymes (assessed using a fluorescence lifetime imaging method with a two-photon laser-scanning microscope) were also found to be overactivated in inflammatory monocytes, activated microglia, and astrocytes of the brain and peripheral CD11b(+) cells of CerTN L15 x LysM tdRFP mice with EAE [195]. In addition, NOX2 deletion or deficiency could prevent EAE induction in female C57BL/6 gp91phox−/− [Nox2 KO mice (gp91Cybbtm1Din/J)] mice (assessment performed using qRT-PCR) [196,197].

### 4.5. Other Markers

Hasseldam et al. [198] reported a loss in mitochondrial membrane potential and oxidative changes (assessed using a spectrophotometric method) in the brains of female Dark Agouti rats with EAE, present approximately 10 days before clinical onset. Other authors, using two-photon imaging, described increased mitochondrial oxidation in oligodendrocytes from MOG-cre mice, CCR2-RFP x CX3CR1-GFP mice, mito-roGFP2-Orp1 mice, and Ai14 reporter mice with EAE [199].

Aheng et al. [200] described a higher severity of EAE in a model of mice lacking the inducible nitric oxide synthase (iNOS) gene, while mice lacking simultaneously uncoupling protein 2 (UCP2) and iNOS genes developed milder EAE.

Johnson et al. [201] described a relationship between the deficiency of nuclear-factor-erythroid-2-related factor 2 (Nrf2) (assessed using qPCR) and a more severe clinical course, a more rapid onset, and a greater percentage of Biozzi ABH mice back-crossed onto Nrf2-KO mice developing EAE after the administration of myelin oligodendrocyte glycoprotein (MOG 35-55), suggesting a role of Nrf2 in modulating the neuroinflammatory response.

Honorat et al. [202] reported an increased expression of xanthine oxidase (assessed using a fluorometric assay) in infiltrating macrophages and microglia of the spinal cord from 8-week-old female SJL/J mice with EAE, which decreased with the preadministration of a potent xanthine oxidase inhibitor, thereby, suggesting a role of xanthine oxidase in the pathogenesis of EAE.

Metabolomic studies in plasma (using ultra-high-performance liquid chromatography-orbitrap-mass spectrometry—UHPLC-Orbitrap-MS) from C57BL/6J EAE mice showed a downregulation in glycerophospholipids and fatty acyls and upregulation in glycolipids, taurine-conjugated bile acids, and sphingolipids, and an increase in NOX activity and MMP9 during disease progression [203]. Increased superoxide anion concentrations and upregulation in NOS3 in the pituitary and adrenal glands were reported, as well as an increase in MDA and GSH levels and in catalase activity (assessed using electron paramagnetic resonance spectroscopy) in adrenal glands of 2-month-old female rats of Dark Agouti with EAE [204]. Plasma concentrations of IgG antibodies with peroxidase [205], oxidoreductase [205], and catalase [206] activities (assessed using spectrophotometry) were increased at different stages of EAE in C57BL/6, Th, 2D2 mice [205], and C57BL/6 mice [206].

### 4.6. Effects of Exposure of Neuronal Cultures to Pathological Products of MS Patients

Vidaurre et al. [207], in a study comparing 13 MS patients with 10 HC, showed that acute exposure of culture neurons to the CSF from MS patients induced oxidative stress and decreased the expression of neuroprotective genes (assessed using RT-PCR and a quantitative lipidomic analysis), which was attributed to an increased content of the ceramides C16:0 and C24:0. Finally, the injection of cultured neurons and oligodendrocytes killed by oxidized phosphatidylcholines obtained from MS lesions induced focal demyelinating lesions with prominent axonal loss in the spinal cord of mice [14].

## 5. Discussion and Conclusions

Many findings point to the possible role of oxidative stress in the pathogenesis of MS. This is supported by a demonstrated increase in markers of oxidation of lipids, proteins, and DNA, and markers of nitrosative stress, together with changes in the activity of enzymes and proteins involved in oxidative stress, both in the brain and/or the spinal cord of MS patients (in samples from autopsies) and in experimental models of MS, mainly in different strains of rodents with EAE (although data from these experimental models could have a low predictive value). Most studies on CSF and blood cells have also shown an increase in several markers of oxidative stress and a decrease in several antioxidant substances in MS patients compared to controls. However, a recent meta-analysis of oxidative stress markers in CSF confirmed only a significant association of CSF MDA levels, but not of other potential markers, with MS [208]. Even though studies of concentrations of markers of oxidative stress in serum/plasma, urine, and other tissues have not shown conclusive results, most studies and the results of a meta-analysis showed an increase in serum/plasma MDA and albumin concentrations in patients diagnosed with MS compared to controls [208]. Similarly, the majority of studies showed a decrease in serum/plasma TAS/TAC and serum/plasma levels of SH groups and an increase in TOS and OSI in patients with MS.

In summary, the main alterations found in studies addressing oxidative stress markers in patients with MS included the following;
Increased markers for lipid peroxidation, protein oxidation, and DNA oxidation.Increased mitochondrial activity.Increased NO nitrotyrosine (therefore, increased nitrosative stress).Increased TOS and OSI and decreased TAS.Decreased iron, copper, and ceruloplasmin.Increased SOD and catalase activities.Decreased GSH levels and normal or decreased GPx activity.Increased NQO1, NOX1, NOX2, and HO-1 activities.Increased myeloperoxidase and peroxiredoxins 1 and 2 activities.Increased endoplasmic reticulum stress proteins.Decreased concentrations of uric acid and related substances.Decreased ascorbate concentrations.

To date, none of the studied variants in genes related to oxidative stress have shown an unequivocal association with MS. Potential reasons for the controversies seen between the results of various studies, both in those addressing biochemical parameters and studies on genes related to oxidative stress, include sample size, differences in sample collection and the methods used, and possibly factors associated with the participants (for example, some studies were not matched by age and/or sex or treatment with disease-modifying therapies). Moreover, in the case of genetic case–control association studies, there was a lack of replication studies.

## 6. Future Directions

We suggest that future studies aiming to establish the possible role of oxidative stress in the pathogenesis of MS should fulfill, at least, the following conditions:Design prospective and multicenter studies with a long-term follow-up period (1 year).The recruitment of a large number of patients diagnosed with MS according to standardized criteria [209], not exposed to any therapy for this disease, and a similar number of age- and sex-matched healthy controls who do not fulfill clinical criteria for the diagnosis of MS and without a family history of MS.Both MS patients and controls involved in such studies should not have obesity or undernutrition, should not suffer from oncologic, acute infectious diseases, kidney, liver, thyroid, or parathyroid disease, have no recent history of traumatism or surgery, and no atypical dietary habits (i.e., diets consisting exclusively of one type of foodstuffs, such as vegetables, and others). They should not use therapy with steroids, diuretics, diphosphonates vitamins, calcium or mineral supplements, or drugs that could affect oxidative stress. In addition, pregnant women should be excluded.It would be desirable to collect plasma/serum and blood cells for the analysis of multiple oxidative stress biomarkers and to obtain blood DNA for genetic studies of genes related to oxidative stress, both in MS patients and controls, at baseline.Patients with MS should undergo periodic clinical evaluations every 3–4 months to evaluate the evolutive type and severity of the disease according to standardized scales such as the EDSS [210].A new collection of plasma/serum and blood cells should be performed for the analysis of multiple oxidative stress biomarkers at the end of the follow-up to evaluate the changes induced by the different treatments used for MS.

## Figures and Tables

**Figure 1 ijms-25-06289-f001:**
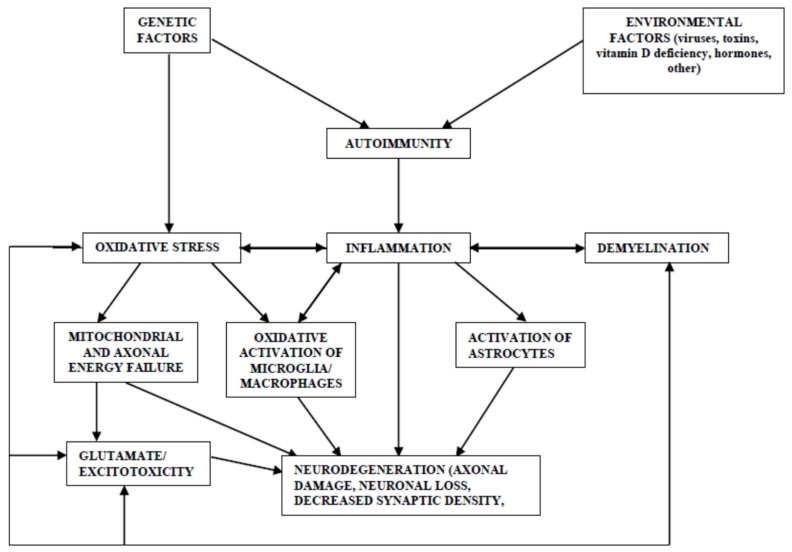
Possible interactions between the different pathogenetic mechanisms in multiple sclerosis.

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
