# Peer review of "Oxidative Stress Markers in Multiple Sclerosis"

_ijms, 2024, doi:10.3390/ijms25126289_

Round 1

Reviewer 1 Report

Comments and Suggestions for Authors

The study is an extensive review considering how markers of oxidative stress relate to the pathology of Multiple Sclerosis. 

Although the authors put a strong effort into resuming data about oxidative stress markers from different tissue types (blood, CSF...), I found the manuscript very hard to follow, particularly because of the very long list of markers reported in the tables. For potential readers interested in a better understanding of the pathogenic pathways of MS, it would be helpful to have data summarised according to hypothetical pathways or by kind of target cells (i. neurons, astrocytes, synapses...). I would suggest hence, organising the results in figures or more synthetic tables according to this approach (or, alternatively, by disease phenotypes) and leaving the complete tables only as supplemental materials for consultation.

Lines 123-124: "similar for the different evolutive types of MS"; this concept should be clarified. What do the authors mean by evolutive type?

Author Response

The study is an extensive review considering how markers of oxidative stress relate to the pathology of Multiple Sclerosis. 

Although the authors put a strong effort into resuming data about oxidative stress markers from different tissue types (blood, CSF...), I found the manuscript very hard to follow, particularly because of the very long list of markers reported in the tables. For potential readers interested in a better understanding of the pathogenic pathways of MS, it would be helpful to have data summarised according to hypothetical pathways or by kind of target cells (i. neurons, astrocytes, synapses...). I would suggest hence, organising the results in figures or more synthetic tables according to this approach (or, alternatively, by disease phenotypes) and leaving the complete tables only as supplemental materials for consultation.

OK. Tables have been moved to Supplementary material, and a figure including the most important pathogenetic pathways of MS has been added

Lines 123-124: "similar for the different evolutive types of MS"; this concept should be clarified. What do the authors mean by evolutive type?

We have changed to “similar for the different clinical subtypes of MS (clinical isolated syndrome -CIS-, relapsing-remitting MS -RRMS-, secondary progressive MS -SPMS-, and primary progressive MS -PPMS)”

Reviewer 2 Report

Comments and Suggestions for Authors

The topic of the manuscript in the form of a review is attractive; but oxidative stress in the mechanism, pathology, and prognosis is not a novel issue. Authors only summation a lot of articles into complicated tables especially table 4, first I commend you for digesting these tables and making a figure as your summary.

In addition, you should illustrate the mechanism of these findings, and make your manuscript easy to read and understand. 

Author Response

The topic of the manuscript in the form of a review is attractive; but oxidative stress in the mechanism, pathology, and prognosis is not a novel issue. Authors only summation a lot of articles into complicated tables especially table 4, first I commend you for digesting these tables and making a figure as your summary.  In addition, you should illustrate the mechanism of these findings, and make your manuscript easy to read and understand. 

Tables have been moved to Supplementary material and a figure including the most important pathogenetic pathways of MS has been added. The most relevant findings for the topic have been described in the text.

Reviewer 3 Report

Comments and Suggestions for Authors

Abstract

Some sentences are long and complex, which may hinder comprehension for certain readers.

The repetition of certain concepts, such as "oxidative stress," could be simplified or varied to avoid monotony and improve the fluidity of the text.

Introduction

Although the aim is mentioned to analyze the role of oxidative stress in multiple sclerosis, it would be useful to have a more direct and concise statement at the beginning to clearly establish the study's focus.

The introduction contains detailed information about genetics and environmental factors related to multiple sclerosis, as well as an explanation of the relationship between oxidative stress and inflammation. This may overwhelm the reader and distract from the study's main focus. It would be beneficial to condense the information and focus on the most relevant aspects for the study.

Although some studies and theories related to multiple sclerosis and oxidative stress are mentioned, a comprehensive literature review is not provided. It would be helpful to include a brief literature review at the beginning to contextualize the topic and highlight the knowledge gap that the study seeks to address.

Point 2: Oxidative stress markers in patients with multiple sclerosis

The enumeration of a large number of enzymes and proteins related to oxidative processes may be confusing for the reader. It would be helpful to group these enzymes and proteins more clearly, possibly categorizing them according to their function or relevance in the context of multiple sclerosis.

The information on oxidative stress markers in cerebrospinal fluid (CSF) could be presented in a more structured manner. For example, you could consider grouping the results of studies according to the type of oxidative stress marker (such as lipid peroxidation markers) to facilitate comparison and understanding.

It would be beneficial to include specific references to the mentioned studies instead of simply listing reference numbers. This would provide the reader with the opportunity to access the original information and evaluate the validity of the presented results. Additionally, ensure consistency in the format of references throughout the text.

2.3. Oxidative stress markers in blood cells

The presentation of results on oxidative stress markers in blood cells could be clearer and more structured. For example, you could group the results according to the type of blood cell (erythrocytes, leukocytes, platelets, etc.) to facilitate comprehension and comparison of the data.

More concise writing: Some parts of the text could be more concise to improve fluency and clarity. For example, you could simplify descriptions of results by removing unnecessary details or repetitions and using more direct and precise wording. This would help maintain the reader's interest and highlight key points more effectively.

4. Data from experimental models of multiple sclerosis

The large amount of information on study results in experimental models of multiple sclerosis may be overwhelming for the reader. It would be helpful to organize the data more clearly and concisely, possibly grouping findings by type of oxidative stress marker or by animal model used.

Some parts of the text could be shorter and more direct, eliminating unnecessary details or repetitions that do not significantly contribute to the understanding of the topic.

The structure of some sentences and paragraphs may be confusing due to the lack of coherence in the presentation of results. It would be beneficial to organize the information more uniformly to facilitate comprehension and comparison of data across different studies.

5. Discussion, conclusions, and future directions

Some parts of the text could be written more clearly and concisely to improve comprehension. For example, the phrase "a new collection of plasma/serum and blood cells for the analysis of multiple oxidative stress biomarkers should be performed" could be simplified to "a new collection of plasma/serum and blood cells should be performed for the analysis of multiple oxidative stress biomarkers." In some cases, the use of verb tenses could be improved to maintain coherence in the text.

Coherence in the presentation of study conditions: The conditions suggested for future studies could be presented more uniformly and structured. For example, they could be organized in a numbered list or with bullet points to facilitate reading and understanding.

Avoiding unnecessary repetitions: Some phrases could be simplified by eliminating unnecessary repetitions.

Great work and congratulations!

Author Response

Abstract

Some sentences are long and complex, which may hinder comprehension for certain readers.The repetition of certain concepts, such as "oxidative stress," could be simplified or varied to avoid monotony and improve the fluidity of the text.

The abstract has been shortened and repetitions of “oxidative stress” reduced.

Introduction

Although the aim is mentioned to analyze the role of oxidative stress in multiple sclerosis, it would be useful to have a more direct and concise statement at the beginning to clearly establish the study's focus.  

We have added some general concepts about oxidative stress and biomarkers of oxidative stress in the introduction section to introduce the reader. We have added a figure including the most important pathogenetic pathways of MS.

The introduction contains detailed information about genetics and environmental factors related to multiple sclerosis, as well as an explanation of the relationship between oxidative stress and inflammation. This may overwhelm the reader and distract from the study's main focus. It would be beneficial to condense the information and focus on the most relevant aspects for the study. Although some studies and theories related to multiple sclerosis and oxidative stress are mentioned, a comprehensive literature review is not provided. It would be helpful to include a brief literature review at the beginning to contextualize the topic and highlight the knowledge gap that the study seeks to address.

We feel that a brief introduction to the various aspects of the etiopathogenic mechanisms of MS could be useful in introducing the position of oxidative stress among them. Subsequently, as we mentioned in the previous point, we have added some general aspects of oxidative stress and its markers

Point 2: Oxidative stress markers in patients with multiple sclerosis

The enumeration of a large number of enzymes and proteins related to oxidative processes may be confusing for the reader. It would be helpful to group these enzymes and proteins more clearly, possibly categorizing them according to their function or relevance in the context of multiple sclerosis.

The list of markers analyzed was previously grouped, but we have added now the specific groups in the Supplementary Tables.

The information on oxidative stress markers in cerebrospinal fluid (CSF) could be presented in a more structured manner. For example, you could consider grouping the results of studies according to the type of oxidative stress marker (such as lipid peroxidation markers) to facilitate comparison and understanding.   

Since the list of biomarkers is very extensive, according to the comments of other reviewers, the tables have been moved to supplementary material for consultation. In the tables, the markers are grouped according to the type of oxidative stress marker. The text has focused on the most relevant findings.

It would be beneficial to include specific references to the mentioned studies instead of simply listing reference numbers. This would provide the reader with the opportunity to access the original information and evaluate the validity of the presented results. Additionally, ensure consistency in the format of references throughout the text.

The results of specific references could be consulted in the tables, now included as Supplementary material.

2.3. Oxidative stress markers in blood cells

The presentation of results on oxidative stress markers in blood cells could be clearer and more structured. For example, you could group the results according to the type of blood cell (erythrocytes, leukocytes, platelets, etc.) to facilitate comprehension and comparison of the data.

the results were grouped according to the type of blood cell in the Supplementary Tables, but according to the marker in the text (some markers were similarly affected in different blood cell types)

More concise writing: Some parts of the text could be more concise to improve fluency and clarity. For example, you could simplify descriptions of results by removing unnecessary details or repetitions and using more direct and precise wording. This would help maintain the reader's interest and highlight key points more effectively.

We have revised the whole text trying to avoid repetitions.

Data from experimental models of multiple sclerosis

The large amount of information on study results in experimental models of multiple sclerosis may be overwhelming for the reader. It would be helpful to organize the data more clearly and concisely, possibly grouping findings by type of oxidative stress marker or by animal model used.Some parts of the text could be shorter and more direct, eliminating unnecessary details or repetitions that do not significantly contribute to the understanding of the topic. The structure of some sentences and paragraphs may be confusing due to the lack of coherence in the presentation of results. It would be beneficial to organize the information more uniformly to facilitate comprehension and comparison of data across different studies.

This part was structured according to the data described in different types of markers in experimental models, mainly in EAE. Finally, some articles describe the effect of exposure of neuronal cultures to pathological products of MS patients

Discussion, conclusions, and future directions

Some parts of the text could be written more clearly and concisely to improve comprehension. For example, the phrase "a new collection of plasma/serum and blood cells for the analysis of multiple oxidative stress biomarkers should be performed" could be simplified to "a new collection of plasma/serum and blood cells should be performed for the analysis of multiple oxidative stress biomarkers." In some cases, the use of verb tenses could be improved to maintain coherence in the text.

The expression was changed as indicated. English was revised by Prof. James McCue, an English native with expertise in editing scientific manuscripts.

Coherence in the presentation of study conditions: The conditions suggested for future studies could be presented more uniformly and structured. For example, they could be organized in a numbered list or with bullet points to facilitate reading and understanding.

Conditions suggested were organized in points a, b, c, d, e, and f.

Avoiding unnecessary repetitions: Some phrases could be simplified by eliminating unnecessary repetitions.

OK, we've tried to reduce repetitions as much as possible

Great work and congratulations!

Reviewer 4 Report

Comments and Suggestions for Authors

Jiménez et al. conducted a narrative review summarizing studies on oxidative stress markers in multiple sclerosis patients, suggesting a potential role of oxidative stress alongside genetic factors, autoimmunity, inflammation, demyelination, and neurodegeneration in the disease's pathogenesis. They emphasized the need for further prospective, multicenter studies with long-term follow-up involving large patient cohorts and controls.

Despite its relevance in the field, this review lacks clarity in providing a definitive and conclusive outcome. The primary issues - major concerns identified include:

Title needs lower case.

Authors need to define better in the introduction what exactly oxidative stress markers are, among other basic modifications. The reader is not adequately “introduced”.

Why didn’t the authors choose to conduct a systematic review with meta-analysis (see this study https://doi.org/10.3390/jcm12196311) Their statement “retrieved 1672 references that were manually selected to include only those strictly related to the topic” introduces bias and results could be viewed skewed. – Why were only 210 references included?

Text descriptions of oxidative stress markers from MS patients in 1) post mortem brains 2) CSF 3) blood cells 4) serum/plasma are adequately described. However, the authors might need to present the healthy control findings in first and second categories more eloquently, as they did with blood cell and serum/plasma category. Of note, line 157 needs rephrase as it starts abruptly with “Table 3. [68-84].” Other fluids such as urine and saliva are somehow underappreciated.

These huge tables could be rearranged into nice schematic graphs (again see https://doi.org/10.3390/jcm12196311)

Line 321: single nucleotide variants (SNVs) à single nucleotide polymorphisms (SNPs)

Line 351 – not 365: experimental allergic encephalomyelitis à Experimental autoimmune encephalomyelitis. Line 385: delete W

Paragraph-Line 408: “13 MS patients with 10 HC” this part should be EAE-exclusive, not MS

With regards to Part 4. EAE is overall poorly structured.

Part 5. Discussion, conclusions, and future directions is probably one of the most essential part in such a review. The authors need to summarize their findings. They just provide a list of criteria for future studies. Where’s the actual discussion about the gigantic tables that probably not going to be read anyways? These tables need a smart grouping and interpretation, not just a plain mentioning.

Additionally, the information about malondialdehyde (MDA), Total Antioxidant Status (TAS), Total Oxidative Status (TOS), and Oxidative Stress Index (OSI) – information that the reader hasn’t encountered yet up until this point – should be moved to the introduction as they are important measurements when discussing oxidative stress markers.

Comments on the Quality of English Language

English grammar and expression need attention throughout the manuscript.

Author Response

Jiménez et al. conducted a narrative review summarizing studies on oxidative stress markers in multiple sclerosis patients, suggesting a potential role of oxidative stress alongside genetic factors, autoimmunity, inflammation, demyelination, and neurodegeneration in the disease's pathogenesis. They emphasized the need for further prospective, multicenter studies with long-term follow-up involving large patient cohorts and controls. Despite its relevance in the field, this review lacks clarity in providing a definitive and conclusive outcome. The primary issues - major concerns identified include:

Title needs lower case.

OK, changed.

Authors need to define better in the introduction what exactly oxidative stress markers are, among other basic modifications. The reader is not adequately “introduced”. 

We have added some general concepts about oxidative stress and biomarkers of oxidative stress in the introduction section to introduce the reader

Why didn’t the authors choose to conduct a systematic review with meta-analysis (see this study https://doi.org/10.3390/jcm12196311) Their statement “retrieved 1672 references that were manually selected to include only those strictly related to the topic” introduces bias and results could be viewed skewed. – Why were only 210 references included?

The reason why we have preferred to carry out a narrative review instead of a systematic review with meta-analysis is because the large number of biomarkers included in the studies that we found in the search of the previous literature is too large, and would require a significant number of analyses and figures. The resulting size of the item would also be excessively large. The final inclusion of 201 references was made after excluding many articles that appeared in the literature search that had no direct relationship with the topic to be addressed, as specified in the introduction.

Text descriptions of oxidative stress markers from MS patients in 1) post mortem brains 2) CSF 3) blood cells 4) serum/plasma are adequately described. However, the authors might need to present the healthy control findings in first and second categories more eloquently, as they did with blood cell and serum/plasma category.

Most studies included a control group. When we mentioned that some parameters were increased or decreased, we were always referring to their comparison with that control group. However, we have changed the wording in the appropriate places to emphasize this issue.

Of note, line 157 needs rephrase as it starts abruptly with “Table 3. [68-84].”

Rephrased to “The results of studies addressing concentrations of oxidative stress markers in blood cells (erythrocytes, leukocytes, peripheral blood mononuclear cells/lymphocytes, and platelets) [68-84] are summarized in Supplementary Table 3”.

Other fluids such as urine and saliva are somehow underappreciated.

These fluids have been the subject of very few studies, and all of them have been discussed in the supplementary tables and in the text.

These huge tables could be rearranged into nice schematic graphs (again see https://doi.org/10.3390/jcm12196311)

Tables have been moved to Supplementary Material and we added a figure.

Line 321: single nucleotide variants (SNVs) à single nucleotide polymorphisms (SNPs) 

OK, changed

Line 351 – not 365: experimental allergic encephalomyelitis à Experimental autoimmune encephalomyelitis. Line 385: delete W

OK, changed and deleted.

Paragraph-Line 408: “13 MS patients with 10 HC” this part should be EAE-exclusive, not MS.

The study mentioned here was performed in neuronal cultures exposed to CSF from patients with MS or controls to evaluate the effect caused by MS (not that marker determinations were made in the CSF). For this reason, it was included in the experimental data

With regards to Part 4. EAE is overall poorly structured.

This part was structured according to the data described in different types of markers in experimental models, mainly in EAE. Finally, some articles describe the effect of exposure of neuronal cultures to pathological products of MS patients

Part 5. Discussion, conclusions, and future directions is probably one of the most essential part in such a review. The authors need to summarize their findings. They just provide a list of criteria for future studies. Where’s the actual discussion about the gigantic tables that probably not going to be read anyways? These tables need a smart grouping and interpretation, not just a plain mentioning.

The main findings were summarized in the first section of the discussion and conclusions (current lines 390-413). The list of proposed criteria for future studies was described immediately after the previous section.

Additionally, the information about malondialdehyde (MDA), Total Antioxidant Status (TAS), Total Oxidative Status (TOS), and Oxidative Stress Index (OSI) – information that the reader hasn’t encountered yet up until this point – should be moved to the introduction as they are important measurements when discussing oxidative stress markers.

We have mentioned the various markers of oxidative stress that were discussed in the introduction

Round 2

Reviewer 4 Report

Comments and Suggestions for Authors

Overall, I am quite satisfied with the changes made. However, I would like to provide some feedback regarding the newly incorporated Figure 1. While it effectively conveys the information, I find its appearance to be quite basic and not particularly attractive. As an additional suggestion for improvement, I recommend creating a second figure where authors can summarize all the removed supplementary material in a more informative and visually appealing schematic representation. Emphasizing multiple sclerosis in this representation would greatly enhance the manuscript's clarity and impact. I believe this addition would significantly contribute to the overall presentation of the research findings. Thank you for considering my suggestion, and I look forward to seeing the final version of the manuscript.

Author Response

REVIEWER 4

Overall, I am quite satisfied with the changes made. However, I would like to provide some feedback regarding the newly incorporated Figure 1. While it effectively conveys the information, I find its appearance to be quite basic and not particularly attractive. As an additional suggestion for improvement, I recommend creating a second figure where authors can summarize all the removed supplementary material in a more informative and visually appealing schematic representation. Emphasizing multiple sclerosis in this representation would greatly enhance the manuscript's clarity and impact. I believe this addition would significantly contribute to the overall presentation of the research findings. Thank you for considering my suggestion, and I look forward to seeing the final version of the manuscript.

OK, A new figure 2 has been added as indicated